

# IASI-derived NH$_3$ enhancement ratios relative to CO for the tropical biomass burning regions

Simon Whitburn[1], Martin Van Damme[1], Lieven Clarisse[1], Daniel Hurtmans[1], Cathy Clerbaux[1,2], and Pierre-François Coheur[1]

[1]Université Libre de Bruxelles(ULB), Atmospheric Spectroscopy, Service de Chimie Quantique et Photophysique CP160/09, Avenue F.D. Roosevelt 50, 1050 Bruxelles, Belgium
[2]LATMOS/IPSL, UPMC Univ. Paris 06 Sorbonne Universités, UVSQ, CNRS, Paris, France.

*Correspondence to:* Simon Whitburn (simon.whitburn@ulb.ac.be)

**Abstract.** Vegetation fires are a major source of ammonia (NH$_3$) in the atmosphere. Their emissions are mainly estimated from "bottom-up" approaches which rely on uncertain emission factors. In this study, we derive new biome-specific NH$_3$ enhancement ratios relative to carbon monoxide (CO), ER$_{NH_3/CO}$ – directly related to the emission factors, from the measurements of the IASI sounder on board the Metop-A satellite. This is achieved for large tropical regions and for a 8-year period (2008–2015). We find substantial differences in the ER$_{NH_3/CO}$ between the studied biomes with calculated values ranging from $4.4 \times 10^{-3}$ to $17 \times 10^{-3}$. For Evergreen Broadleaf Forest these are typically 75-100% higher than for Woody Savanna and Savanna biomes. This variability is attributed to differences in fuel types and size and is in line with previous studies. The analysis of the spatial and temporal distribution of the ER$_{NH_3/CO}$ also reveals a (sometimes large) within-biome variability. On a regional level, Woody Savanna shows for example a mean ER$_{NH_3/CO}$ for the region of Africa South of the Equator which is 50-100% lower than in the other five studied regions, probably reflecting regional differences in fuel type and burning conditions. The same variability is also observed on a yearly basis with a peak in the ER$_{NH_3/CO}$ observed for the year 2010 for all biomes. These results highlight the need for the development of dynamic emission factors that better take into account local variations in fuel type and fire conditions. We also compare the IASI-derived ER$_{NH_3/CO}$ with values reported in the literature, usually calculated from ground-based or airborne measurements. We find a general underestimation over the referenced ER$_{NH_3/CO}$ of about 40% for Woody Savanna and Savanna and up to a factor 1.5-4 for Evergreen Broadleaf Forest and Cropland. Beyond a possible overestimation of the ER$_{NH_3/CO}$ in the literature, the observed differences could also be related to various factors including instrumental limits, bias in the retrieval of the NH$_3$ columns, parameterization in the calculation of the ER$_{NH_3/CO}$ or accumulation of CO in the studied regions during the fire period.

## 1 Introduction

Vegetation fires contribute significantly to the global budget of many trace gases and aerosols in the atmosphere (Langmann et al., 2009). Carbon dioxide (CO$_2$) emissions from biomass burning are for example estimated to about 2-4 Pg C year$^{-1}$ against 7.2 Pg C year$^{-1}$ from fossil fuel combustion (Bowman et al., 2009). For carbon monoxide (CO), the contribution to the total budget could even reach more than 50% (Crutzen and Andreae, 1990; van der Werf et al., 2004, 2010). Besides





carbon, vegetation fires also emit large amounts of reactive nitrogen species, among which ammonia ($NH_3$). With a contribution estimated to be about 13% (Galloway et al., 2004) of the total emissions, biomass burning is believed to be the second most important source of $NH_3$ after agriculture. From previous studies, it has been shown that biomass burning could affect significantly $NH_3$ concentrations in the atmosphere, especially in the tropics but also at higher latitudes (e.g., Bouwman et al.,

1997; Coheur et al., 2009; Adon et al., 2010; Alvarado et al., 2011; Shephard et al., 2011; Adon et al., 2013; R'Honi et al., 2013; Whitburn et al., 2015, 2016a; Benedict et al., 2017; Warner et al., 2017). Excess $NH_3$ in the environment is of great concern since it is responsible for many environmental issues such as eutrophication of terrestrial and aquatic ecosystems, soil acidification and loss of plant diversity (Aneja et al., 2001; Erisman et al., 2007). As the dominant alkaline species in the atmosphere, $NH_3$ rapidly combines with acid gases such as sulfuric acid ($H_2SO_4$), nitric acid ($HNO_3$) and hydrochloric acid

(HCl) resulting in the formation of secondary aerosols that are in turn impacting climate and human health (Bouwman et al., 1997; Aneja et al., 2001; Sutton et al., 2011; Behera et al., 2013; Lelieveld et al., 2015).

    Until recently, most models of fire emissions were based on "bottom-up" approaches which rely on an estimation of the total burned biomass (BB, kg) combined with biome-specific emission factors (EFs), expressed as the mass of pollutant emitted per kilogram of BB (g kg$^{-1}$ BB). Despite the numerous studies achieved in the past decades (e.g., Sinha et al., 2003; Yokelson

et al., 2003; van der Werf et al., 2010; Wooster et al., 2011; Smith et al., 2014), the uncertainty on all parameters of these models remain large. This is especially true for EFs, which have typical uncertainty of the order of 20-30% for frequently measured species (e.g. CO, $CO_2$) and up to 100% for species such as $NH_3$ which are not so well monitored (Langmann et al., 2009; Akagi et al., 2011). An accurate determination of the EFs is challenging partly because of the existence of a within-biome spatial and seasonal variability (van Leeuwen and van der Werf, 2011; Yokelson et al., 2011; Meyer et al., 2012; Mebust and Cohen, 2013;

van Leeuwen et al., 2013; Castellanos et al., 2014; Schreier et al., 2014a). This variability is attributed to differences in fuel type and burning conditions, the latter being itself controlled by climate, weather, moisture content, topography and fire practices (Andreae and Merlet, 2001; Korontzi et al., 2003; Yokelson et al., 2011; van Leeuwen and van der Werf, 2011; Castellanos et al., 2014). For nitrogen compounds, another main factor controlling the EFs is the Nitrogen content of the fuel (Andreae and Merlet, 2001; Jaffe and Wigder, 2012; Castellanos et al., 2014). Because it is generally not known to what extent EFs are based

on a representative sample of a specific vegetation type (van Leeuwen and van der Werf, 2011; Castellanos et al., 2014), the spatial and temporal variability in the EFs is usually not taken into account in the bottom-up approaches where EFs are taken from compilations of airborne and local measurements or from small fires burned under laboratory conditions (e.g., Andreae and Merlet, 2001; Akagi et al., 2011).

    With their excellent spatial and temporal coverage, hyperspectral sounders on board satellites, directly measuring tropo-

spheric concentration of trace gases in the atmosphere, offer a unique opportunity to determine EFs more accurately and to capture their variability in time and space. Nowadays, the focus was principally on CO, nitrogen dioxide ($NO_2$) and aerosols (e.g., Pechony et al., 2013; Castellanos et al., 2014; Ichoku and Ellison, 2014; Mebust and Cohen, 2014; Schreier et al., 2014a, b). A recent study was also dedicated to formic acid (HCOOH) (Pommier et al., 2017). Until now, less attention has been given to $NH_3$ (Coheur et al., 2009; Alvarado et al., 2011; R'Honi et al., 2013; Luo et al., 2015). In this paper we derive biome-specific

$NH_3$ enhancement ratios relative to CO ($ER_{NH_3/CO}$, also known as normalized excess mixing ratios) and relate them to EFs



(see section 2.2), over large tropical fire regions and long periods using the measurements of the Infrared Atmospheric Sounding Interferometer (IASI). The use of IASI is particularly suitable here because of its exceptional sampling (compared to other similar instruments, such as the Tropospheric Emission Spectrometer (TES) (Shephard et al., 2011)) and to our knowledge, it is the first time such a study focusing on biomass burning ERs is carried out at this scale for $NH_3$. Section 2 hereafter briefly

5    describes the datasets used and introduces the methodology for calculating the enhancement ratios. It also motivates the selection of the regions studied. The results from our analyzes are presented and discussed in section 3, which is further divided in two main parts. The first part analyzes the variability of $ER_{NH_3/CO}$ between and within the different biomes (an extensive comparison with ERs reported in the literature is also provided) while the second analyzes the interannual and seasonal evolution of $ER_{NH_3/CO}$. Summary and conclusion are given in section 4.

## 2    Dataset and Method

### 2.1    Instruments and data products

IASI is a nadir-looking high resolution Fourier Transform Spectrometer on board the polar orbiting sun-synchronous Metop (Meteorological Operational) satellites. The two first IASI sounders were launched in 2006 and 2012 (Metop-A and -B, respectively). A third instrument is scheduled for launch in 2018 and will ensure at least 18 years of consistent measurements

(2006–2023). IASI covers the entire globe twice daily (09:30 and 21:30 local time when crossing the equator) with a relatively small elliptical footprint on the ground varying from $12 \times 12$ km (at nadir) up to $20 \times 39$ km (off nadir); depending on the viewing angle (Clerbaux et al., 2009). Its large and continuous spectral coverage of the thermal infrared band region (645–2760 $cm^{-1}$), its medium spectral resolution (0.5 $cm^{-1}$ apodized) and its low instrumental noise ($\sim$0.2 K at 950 $cm^{-1}$ and 280 K) make it an invaluable instrument for monitoring atmospheric composition (Clerbaux et al., 2009). CO is retrieved from IASI

measurements using the FORLI (Fast Optimal Estimation Retrievals on Layers for IASI) software (Hurtmans et al., 2012). The retrieval of $NH_3$ is based on a new and flexible retrieval algorithm, which relies on the calculation of a so-called Hyperspectral Range Index (HRI) and subsequent conversion to a $NH_3$ total column (molec.$cm^{-2}$) using a neural network (Whitburn et al., 2016b). For a detailed description of the retrieval methods and parameters, we refer the reader to Whitburn et al. (2016b) for $NH_3$ and Hurtmans et al. (2012) for CO. The validation of FORLI-CO profiles and columns have shown good agreement

overall using in-situ, aircraft and satellite observations (Pommier et al., 2010; De Wachter et al., 2012; Kerzenmacher et al., 2012; George et al., 2015). For $NH_3$ columns, the validation has started but is more difficult considering the important spatial and temporal variability of $NH_3$ and the scarcity of correlative ground- and airplane-based measurements in many regions of the world (Van Damme et al., 2015). Two studies, based on a previous $NH_3$ retrieval algorithm also using the HRI but relying on a two-dimensional look-up tables for the conversion into a $NH_3$ total column (molec.$cm^{-2}$) (Van Damme et al., 2014), have

shown fair agreements between IASI-$NH_3$ observations and other measurements (Van Damme et al., 2015; Dammers et al., 2016).

This work makes use of 8 years (2008–2015) of daily global $NH_3$ and CO total columns (molec.$cm^{-2}$) from the measurements of IASI on board Metop-A. Only daytime satellite observations have been considered as they usually show a better





sensitivity, especially to $NH_3$. We also have assumed a similar sensitivity for IASI to $NH_3$ and CO in the lower layers of the atmosphere, which is not expected to introduce a significant bias in the studied regions due to a generally positive thermal contrast prevailing during daytime (Clarisse et al., 2010; Van Damme et al., 2014). A more important bias may result from the use of a unique vertical profile shape in the retrieval scheme of $NH_3$ total columns which is therefore not representative of the large variety of profiles observed above biomass burning plumes. Whitburn et al. (2016b) have calculated that the use of an alternative profile could affect the retrieved column up to 50%. This is important to keep in mind for the analyses presented next.

We also used in support to the selection of the studied regions and the $NH_3$ and CO columns, active fires detection data and nitrogen dioxide ($NO_2$) total columns (molec.cm$^{-2}$). Detected active fires are taken from the Global Monthly Fire Location Product (MCD14ML, Level 2, Collection 5) developed by the University of Maryland from the measurements of the MODerate resolution Imaging Spectroradiometer (MODIS) on board the NASA Terra and Aqua satellites (Justice et al., 2002; Giglio et al., 2006). Active fires are monitored at a resolution of $1 \times 1$ km$^2$ with fires as small as 100 m$^2$ detected. $NO_2$ total columns are retrieved from the measurements of the Global Ozone Monitoring Experiment (GOME-2) also on board the Metop satellites and working in the UV-Vis spectral band region (Valks et al., 2011).

## 2.2 Enhancement ratios

From the IASI $NH_3$ and CO total columns (molec.cm$^{-2}$), we have derived $NH_3$ enhancement ratios relative to CO ($ER_{NH_3/CO}$) defined as the ratio of the number of emitted molecules of $NH_3$ (here the $NH_3$ total column) over the emitted molecules of the reference species CO (here the CO total column) (Andreae et al., 1988; Lefer et al., 1994; Hobbs et al., 2003). The choice of CO as reference species is here particularly suitable as it is a dominant species emitted by fires and has a lifetime of several weeks in the free troposphere. One main advantage of the ERs compared to the EFs is that ERs calculation only requires simultaneous measurements of the studied ($NH_3$) and the reference species (CO) while EFs calculation requires fuel information that are not always available or completely reliable (Andreae and Merlet, 2001). In fire plumes, ERs can be estimated following (Goode et al., 2000; R'Honi et al., 2013):

$$ER_{NH_3/CO} = \frac{[NH_3]_{smoke} - [NH_3]_{ambient}}{[CO]_{smoke} - [CO]_{ambient}} \qquad (1)$$

When a lot of measurements are available, which is often the case for IASI-derived measurements owing to its excellent spatial and temporal resolution, average $ER_{NH_3/CO}$ can be estimated from the slope of the linear regression of $NH_3$ versus CO (Andreae and Merlet, 2001; Coheur et al., 2009). The ERs can also be derived directly from the EFs by multiplying the ratio $EF_{NH_3}/EF_{CO}$ with the ratio of the molar masses $M_{CO}/M_{NH_3}$ (Andreae and Merlet, 2001). This will be used here to convert the reported EF values from ground-based and airborne studies into ERs in order to allow the comparison with our IASI-derived $ER_{NH_3/CO}$.





## 2.3 Selection of the areas and biomes and calculation of the $ER_{NH_3/CO}$

One of the key steps in this study is the selection of the areas of interest for the calculation of the $ER_{NH_3/CO}$. To be relevant, $ER_{NH_3/CO}$ need to be calculated for areas were fires are the dominant source of emissions of $NH_3$ and CO. The selection has been done on a pixel basis. We have first calculated the linear regressions, globally on a $1° \times 1°$ grid, between the monthly

means of the couples $NH_3$–CO total columns (molec.cm$^{-2}$), $NH_3$–$NO_2$ total columns, and $NH_3$ total columns–number of active fires (#fires). We have next selected the pixels for which a correlation coefficient (r) higher than 0.3 was found for the three couples of regression ($NH_3$–CO, $NH_3$–$NO_2$ and $NH_3$–#fires). These are shown in Fig. 1 (colored pixels) and constitute the areas considered for the calculation of the $ER_{NH_3/CO}$. Pixels with a r higher than 0.3 for the considered couple but not for (at least) one of the two other couples are shown in gray. The idea behind this selection procedure is that a good correspondence

between the monthly means of $NH_3$, CO and $NO_2$ total columns provides an indication of a dominant contribution of the fires to their emissions, since biomass burning is indeed the only major common source of emissions of these three species. A significant positive correlation between the $NH_3$ total columns and the detected number of fires adds an additional argument in favor of the contribution of fires and ensures keeping only those areas that are close to the source of emissions, making the comparison with ground-based and airborne derived EFs and ERs easier. In general, the largest correlations are found between

$NH_3$ and CO total columns (Fig. 1, top panel) with correlation coefficients ranging from about 0.6-0.7 up to 0.9 in Africa South of the Equator and Indonesia. The fact that these two species are measured simultaneously from IASI could contribute to this. For the two other couples ($NH_3$–$NO_2$ and $NH_3$–#fires), the correlation coefficients are in the range 0.3–0.8. Note that in general, significant positive correlations between $NH_3$ and $NO_2$ (Fig. 1, middle panel) are only found close to the source of emissions due to the relatively short lifetime of $NO_2$ (of a few hours, Schreier et al., 2014b). With a lifetime of typically 12–36

20  hours in the studied regions (Dentener and Crutzen, 1994; Aneja et al., 2001; Whitburn et al., 2015, 2016a), $NH_3$ is more likely to be transported over longer distances. This can be seen on the $NH_3$ – CO correlation map where positive correlations are also found over seas downwind of the source areas.

For each of the selected pixels, we have next calculated an $ER_{NH_3/CO}$ per year between 2008 and 2015 from the slope of the linear regression between $NH_3$ and CO retrieved columns (molec.cm$^{-2}$). To take into account the $NH_3$ and CO columns most

likely related to fire emissions, we have only considered IASI measurements located within 50 km from a fire. We have also included a quality filter on the $NH_3$ and CO measurements: only total columns with a relative error lower than 100% for $NH_3$ and 25% for CO were retained for the regression. Finally, as a post-filtering, we have only kept for the analysis the $ER_{NH_3/CO}$ for which the linear regressions between $NH_3$ and CO columns show a correlation coefficient larger than 0.3 and for which we have more than 10 measurements. The impact of these pre- and post-filters on the calculated $ER_{NH_3/CO}$ are discussed in

section 3.1. An example of a linear regression between $NH_3$ and CO for one of the selected pixels (Evergreen Broadleaf Forest (EBF) in Indonesia) is given in Fig. 2.

For this study, we focus on the four dominant biomes in the selected pixels. These were identified using the MODIS *Land Cover Type* product (MCD12Q1) with the 17-class International Geosphere-Biosphere Program classification (IGBP) (Friedl et al., 2010) (Fig. 3). The four selected classes are 1) the Evergreen Broadleaf Forests (EBF), 2) the Woody Savannas (WS), 3)





the Savannas (S) and 4) the Crops together with the Crop and Natural Vegetation Mosaic (C+CNVM), here denoted C. Fig. 3 also shows the distribution of the mean yearly $ER_{NH_3/CO}$ averaged over the time period 2008–2015 for the selected pixels. A first analysis of the distribution of the $ER_{NH_3/CO}$ reveals a variability between the four biomes, especially in Africa North of the Equator and in Central South America where a gradient is observed between EBF and WS and between EBF and S,

respectively, with higher $ER_{NH_3/CO}$ found for EBF. A clear gradient is observed as well in Africa South of the Equator from the northwest to the southeast.

The pixel-based $ER_{NH_3/CO}$ have next been grouped by biome to analyze their regional and temporal variability. In addition, to facilitate the study of the spatial distribution of the $ER_{NH_3/CO}$, we have defined six main regions which include the majority of the pixels of interest (see Fig. 1). Two are in Africa, one North (AFR.NEQ.) and one South (AFR.SEQ.) of the Equator. One

corresponds to the central part of South America (S.AM.). A second region in America (C.AM.) is located north to the S.AM. region and includes the region around the Gulf of Mexico, Central America, Colombia and Venezuela. The two last regions are in Asia; one is for South-East Asia (SE. ASIA) and the second is for Indonesia (INDO.).

## 3    Results and discussion

### 3.1    $ER_{NH_3/CO}$ spatial analysis

We analyze here the spatial and biome variability in the $ER_{NH_3/CO}$. For each of the 4 biomes (EBF, WS, S, C) and each of the 6 regions a mean $ER_{NH_3/CO}$ was obtained by averaging all yearly pixel based $ER_{NH_3/CO}$ in the time period 2008 and 2015 (Fig. 4, solid error bars). Mean $ER_{NH_3/CO}$ for the 6 regions globally are shown as well (horizontal lines). Overall, the highest mean $ER_{NH_3/CO}$ is found for EBF ($9.8 \times 10^{-3}$) while S and WS show mean $ER_{NH_3/CO}$ about 75-100% lower with values of $5.9 \times 10^{-3}$ and $7.0 \times 10^{-3}$, respectively. The larger $ER_{NH_3/CO}$ for Evergreen Broadleaf Forests (EBF) compared to

Woody Savannas (WS) and Savannas (S) is in agreement with previous studies (e.g. Andreae and Merlet, 2001; Akagi et al., 2011; Yokelson et al., 2011) and is mainly attributed to differences in fuel size and density: EBF, characterized by dense fuel, is indeed dominated by smoldering combustion, which emits more reduced or incompletely oxidized products (among them $NH_3$ and CO) than grassland (van Leeuwen and van der Werf, 2011). One should note, however, that Kaiser et al. (2012) reported higher $ER_{NH_3/CO}$ for Savannas than for tropical forests. For Crop (C), the mean $ER_{NH_3/CO}$ ($9.0 \times 10^{-3}$) is close to

the $ER_{NH_3/CO}$(EBF) but is more difficult to interpret because the biome probably includes different types of fuel. Figure 5, representing the cumulative frequency of the pixel-based yearly $ER_{NH_3/CO}$ per biome, also shows the biome-trends in the $ER_{NH_3/CO}$. EBF and C have for example about 35% of the calculated $ER_{NH_3/CO}$ above 0.01 while this value corresponds to only about 10–15% for S and WS. These differences in the $ER_{NH_3/CO}$ between biomes are, however, not necessarily found when looking at the average $ER_{NH_3/CO}$ at the region scale. For Central America (C.AM.) and South-East Asia (SE.ASIA) in

particular, the differences between $ER_{NH_3/CO}$ are low (of the order of 5–10%). For South America (S.AM.), $ER_{NH_3/CO}$(EBF) is about twice higher than $ER_{NH_3/CO}$(S) but close to $ER_{NH_3/CO}$(WS) (within 10%).

When comparing the $ER_{NH_3/CO}$ by biome between the six regions in Figure 4 (solid error bars), we find good agreements but also large differences, in line with what has already been reported by, for example, van Leeuwen and van der Werf (2011);

(c) Author(s) 2017. CC-BY 3.0 License.





van Leeuwen et al. (2013); Castellanos et al. (2014). Among the most noticeable differences, we find $ER_{NH_3/CO}$(EBF) up to a factor two higher for the region of Africa North of the Equator (AFR.NEQ.) than for the S.AM., C.AM. and SE.ASIA regions. Similarly, a large variability in the $ER_{NH_3/CO}$ is found for the WS and S biomes ranging between about $5 \times 10^{-3}$ for the region(s) of AFR.SEQ. (and S.AM.) for WS (S) and $10 \times 10^{-3}$ for C.AM. (and S.AM.) for S (WS). A variability is also

observed, but to a lesser extent, for the C biome ranging between $7.3 \times 10^{-3}$ for SE.ASIA and $10.5 \times 10^{-3}$ for C.AM. Note that this intra-biome variability is also found within a given region, as observed in Fig. 3 and as evidenced by the sometimes large standard deviation (std) associated with the mean $ER_{NH_3/CO}$ (e.g. EBF in the AFR.NEQ. region with a std of about 0.01). As mentioned in section 1, these differences can be explained by changes in the fuel type (size and density) but also the climate, weather, topography, moisture and N content, and fire practices. In addition for EBF, different regional deforestation practices

could also lead to variation in the $ER_{NH_3/CO}$ (van Leeuwen and van der Werf, 2011). It should finally be mentioned that for the AFR.NEQ. region, the measured $NH_3$ columns at the end of the fire period probably originate from the combination of both biomass burning emissions and another source, possibly agriculture as suggested in Whitburn et al. (2015); this might therefore introduce a bias in the $ER_{NH_3/CO}$. Overall, these results clearly highlight the need for developing new regional-dependent EFs, in order to improve the representativeness of estimations from bottom-up inventories.

The comparison of the IASI-derived $ER_{NH_3/CO}$ with the values reported in the literature from ground-based or airborne studies (see Table 1) shows a general overestimation of the latter (or an underestimation from IASI), especially for the biomes EBF and C where a factor 1.5-4 difference is observed. The only exception is for Kaiser et al. (2012) for EBF for which a better correspondence (within 50% difference) is found. Note that the largest difference for EBF is with an $ER_{NH_3/CO}$ reported for tropical dry forest (Yokelson et al., 2011), but the latter is likely not representative for the complete Evergreen Broadleaf Forest

class. For S and WS, the agreement is much better (with a maximum IASI underestimation of about 40%) except for Andreae and Merlet (2001), Bertschi et al. (2003) and Kaiser et al. (2012) reporting $ER_{NH_3/CO}$ up to a factor 4 higher. Note that, in Bertschi et al. (2003) $ER_{NH_3/CO}$ are derived from smoldering logs for which higher values are logically expected. Note also that for WS, the $ER_{NH_3/CO}$ are compared here to values reported for Savannas, which are usually included with S into the same biome. While an overestimation of the average $ER_{NH_3/CO}$ (or $EF_{NH_3}$) in the literature is possible, other reasons are likely to

play a role. First, the differences with the IASI-derived $ER_{NH_3/CO}$ could also be (at least partly) explained by the consideration in our work of IASI measurements within 50 km of an active fire, while ground and airborne measurements are done in the direct vicinity of the fire. Second, another possible reason might lie in the difficulty for MODIS to detect smoldering fires, causing the IASI-derived $ER_{NH_3/CO}$ to reflect preferentially the flaming phase of the vegetation fires. Third, an accumulation of CO in the region during the fire period (due to its much longer lifetime compared to $NH_3$) might introduce a bias in the

calculated $ER_{NH_3/CO}$. Finally, the differences with the reported $ER_{NH_3/CO}$ could also be due to the chosen methodology for the calculation of the $ER_{NH_3/CO}$. To verify this, we have recalculated mean biome-specific $ER_{NH_3/CO}$ for the six regions (not shown) by varying one by one the pre- and post-filters considered before (see section 2.3). We have performed four tests: 1) with a maximum distance of the $NH_3$ total column to a detected fire of 30 km and 2) 100 km (instead of 50 km), 3) with a maximum error on the $NH_3$ total column of 75% (against 100%) and 4) by filtering out the $ER_{NH_3/CO}$ for which the linear

regressions between $NH_3$ and CO columns show a correlation coefficient (r) of the linear regression lower than 0.6 (instead





of 0.3). We find a very limited impact of the distance to a fire and the error on the $NH_3$ column allowed, with differences of only about 3–8% (interestingly, an increase –decrease– of the tolerance on the maximum distance to a fire systematically slightly decrease –increase– the mean $ER_{NH_3/CO}$). In contrast, an increase to 0.6 of the threshold on the correlation coefficient introduces a large increase in the mean $ER_{NH_3/CO}$ of about 25-35% (and up to 40% for WS in the AFR.SEQ. region). Taking

into account this increase, we find mean $ER_{NH_3/CO}$ closer (but still below) to what is reported in the literature, especially for WS and S. The agreement would become even better if we consider in addition a possible bias due to the use of a non-representative $NH_3$ vertical profile considered for the retrieval of the $NH_3$, as mentioned in section 2.1. Note that despite the impact of the pre- and post-filters chosen, the analysis on the regional and inter-biome variability in the $ER_{NH_3/CO}$ remains valid.

At a regional level (all biomes combined), a comparison with the satellite-derived $ER_{NH_3/CO}$ based on TES measure-

ments (Luo et al., 2015) shows again higher values of about 50% compared to our calculated $ER_{NH_3/CO}$ (Table 1). However, the calculation of $ER_{NH_3/CO}$ in Luo et al. (2015) are derived from seasonal averages in a $2° \times 4°$grid and might therefore include other sources of emissions in the studied regions, such as agriculture and livestock. Luo et al. (2015) also derived $ER_{NH_3/CO}$ from simulations of the GEOS-Chem global chemical transport model. A good agreement is found between IASI and GEOS-Chem for the regions of AFR.NEQ. and S.AM. with $ER_{NH_3/CO}$ in the range of values calculated for North-Central

Africa and South America. For South-Central Africa in contrast, Luo et al. (2015) reported $ER_{NH_3/CO}$ values of about 3 times higher compared to our AFR.SEQ. region.

## 3.2   $ER_{NH_3/CO}$ interannual and seasonal variability

In this second part, we focus our analysis on the temporal variability in the $ER_{NH_3/CO}$. Fig. 6 shows the mean $ER_{NH_3/CO}$ averaged by biome and by year (2008–2015). The solid line represents the 8-years average for each biome. We find an interannual

variability in the mean $ER_{NH_3/CO}$ up to a factor two for the four studied biomes. Interestingly, the minimum $ER_{NH_3/CO}$ is found in 2012 for all biomes. Similarly the highest mean $ER_{NH_3/CO}$ is observed in 2010 for all biomes (especially marked for EBF) except for S for which the maximum is found in 2008 (despite an $ER_{NH_3/CO}$ for 2010 also above the 8 years average). When analyzing the variability in the yearly averaged $ER_{NH_3/CO}$ for each region separately (Fig. 7), we find that the high mean $ER_{NH_3/CO}$ of 2010 for EBF is exclusively carried by the AFR.NEQ. region with a mean $ER_{NH_3/CO}$ of $40 \times 10^{-3}$ (against about

$15 \times 10^{-3}$ for the other years in the region). For the WS biome, the peak of 2010 is mainly due to the S.AM., AFR.NEQ. and SE.ASIA regions with an $ER_{NH_3/CO}$ about a factor 1.5–2 higher compared to the other years. This important variability in the $ER_{NH_3/CO}$ are probably due to differences in the burning conditions from one year to another. One possible reason to explain the high mean $ER_{NH_3/CO}$ for 2010 in the different regions is the El Niño Southern Oscillation (ENSO) event that occurred that year and that was responsible for severe droughts and increased fire activity in the studied regions (Whitburn et al., 2015).

This is however probably not sufficient to explain the 3 times increase for EBF for 2010 in the AFR.NEQ. region but no clear evidence of other processes influencing the $ER_{NH_3/CO}$ were found for that year. Surprisingly, the same increase in the $ER_{NH_3/CO}$ are not observed for the year 2015, which was the strongest El Niño year since 1997 (Chisholm et al., 2016). For WS, high $ER_{NH_3/CO}$ are in addition observed for 2011 for South and Central America (S.AM. and C.AM.). However, this has small impact on the global yearly $ER_{NH_3/CO}$, which is mainly driven by the two regions in Africa (AFR.NEQ. and AFR.SEQ.),





representing about 33% and 40% of all the calculated $ER_{NH_3/CO}$ for WS, respectively (see Fig. 4). For the S biome, the yearly $ER_{NH_3/CO}$ is largely dominated by the AFR.SEQ. and the S.AM. (47% and 33% of the $ER_{NH_3/CO}$, respectively). Despite being not as pronounced on the global yearly $ER_{NH_3/CO}$ (Fig. 6) as for other biomes, we observe a peak in the $ER_{NH_3/CO}$ for 2010 for the AFR.NEQ., the S.AM. and the C.AM. regions. Note that here the AFR.NEQ. and C.AM. regions also show high $ER_{NH_3/CO}$

for the year 2015 which tends to support the hypothesis of the influence of El Niño on the $ER_{NH_3/CO}$. Finally, the C biome is mainly driven by the AFR.NEQ. region (60% of all $ER_{NH_3/CO}$) showing as well the highest mean $ER_{NH_3/CO}$ in 2010 and 2015. This important variability in the $ER_{NH_3/CO}$ in time and space highlights here again the importance to use dynamic EFs datasets in the fire emission inventories in order to better take into account the local fire conditions.

Finally we investigate the temporal variability in the $ER_{NH_3/CO}$ from a seasonal perspective. For this, we have calculated for

each pixel selected in section 2.3 a separate $ER_{NH_3/CO}$ for the early and for the late fire season. The separation early/late fire season has been done by analyzing the daily time series of the number of fires between 2008 and 2015 for each region and biome studied (not shown). The results are shown in Fig. 4 (dashed error bars). In general, we do not find systematic difference in the $ER_{NH_3/CO}$ between the early and late fire season except for the AFR.NEQ. region, for which the late $ER_{NH_3/CO}$ are higher by about 15–40% for the four biomes. This is in agreement with the hypothesis made in section 3.1 of the presence

of a secondary source of $NH_3$ (possibly agriculture) towards the end of the fire season. The same difference in the $ER_{NH_3/CO}$ was also observed by Luo et al. (2015) who found a 60% increase between the beginning and the end of the fire season for North-Central Africa. Note finally that the early and late fire season $ER_{NH_3/CO}$ are generally close to the corresponding yearly $ER_{NH_3/CO}$ (within 10-30%) which tend to support our methodology for the calculation of the ERs.

## 4   Conclusions

In this work, we have calculated biomass burning $ER_{NH_3/CO}$ over large tropical regions and a 8-year period of IASI satellite measurements for four different biomes, namely Evergreen Broadleaf Forest (EBF), Woody Savannas (WS), Savannas (S) and Cropland (C). Such a study had, to our knowledge, never been performed at this level (in time and space) for $NH_3$. Overall, the results have shown the great potential of IASI for calculating time and space dependent ERs. The $ER_{NH_3/CO}$ have been calculated on a pixel basis from the slope of the linear regression of $NH_3$ versus CO total columns (molec.cm$^{-2}$) retrieved

from IASI measurements. On average, the biomes EBF and C showed $ER_{NH_3/CO}$ about 75-100% higher than WS and S and this was attributed to differences in fuel size and density, affecting the fraction of smoldering combustion. The biome-specific $ER_{NH_3/CO}$ have next been grouped by regions and by year to analyze their spatial and temporal variability. We found an important variability both in time and space for all situations but especially for WS showing a mean $ER_{NH_3/CO}$ about 50-100% lower in Africa South of the Equator (AFR.SEQ.) than in the five other regions, possibly due to local differences in fuel type and

burning conditions. Another interesting feature was the high mean $ER_{NH_3/CO}$ of $17 \times 10^{-3}$ (and up to a factor two higher than for the other studied regions) calculated for Africa North of the Equator (AFR.NEQ.) for EBF. We have tentatively explained this high value by the presence of another source of emissions than biomass burning towards the end of the dry season. This was supported by our analysis of the seasonal dependence in the $ER_{NH_3/CO}$, showing $ER_{NH_3/CO}$ systematically higher for the late



fire season in the AFR.NEQ. region (for the four biomes) than for the beginning of the fire period. The interannual variability in the $ER_{NH_3/CO}$ was also found important (up to a factor 2), with a peak for 2010 for each biome, possibly related to the severe droughts that have occurred that year in the studied regions consequently to an important El Niño event. The important variability of the $ER_{NH_3/CO}$ both in time and space clearly shows the need for developing dynamic datasets of EFs which better

take into account the fuel type and fire conditions.

In comparison to the values reported in the literature, mainly from ground-based and airborne studies, we found a general underestimation of the mean IASI-derived $ER_{NH_3/CO}$ of about 40% for S and WS and up to a factor 1.5-4 for the EBF and C biomes. These differences may be explained by various factors including 1) the parametrization (pre- and post-filtering of the data) considered for the calculation of the $ER_{NH_3/CO}$, 2) a bias towards the flaming phase due to the selection of IASI

observations close to MODIS active fires (less sensitive to the smoldering phase) and 3) a possible accumulation of CO in the region during the fire season, introducing a low-bias in the IASI-derived $ER_{NH_3/CO}$. The comparison with the recent work of Luo et al. (2015) who derived mean $ER_{NH_3/CO}$ from TES measurements tends to confirm a low-bias of the IASI-derived $ER_{NH_3/CO}$.

*Data availability.* The IASI FORLI CO and NH$_3$ Neural Network data used in this work are publicly available for all users through the

French AERIS database (http://iasi.aeris-data.fr/)

*Acknowledgements.* IASI has been developed and built under the responsibility of the Centre National d'Études spatiales (CNES, France). It is flown on board the Metop satellites as part of the EUMETSAT Polar System. The IASI L1 data are received through the EUMETCast near real-time data distribution service. We thank the NASA for providing MODIS fire radiative power data. We also acknowledge the use of the MODIS global land cover map. We thank EUMETSAT for the use of the operational EUMETSAT O3MSAF NO$_2$ product. The algorithm for

the retrieval of the NO$_2$ total columns used in this work has been developed in the context of the *Satellite Application Facility on Ozone and Atmospheric Chemistry Monitoring* (O3M-SAF). The research in Belgium was funded by the F.R.S.-FNRS and the Belgian State Federal Office for Scientific, Technical and Cultural Affairs (Prodex arrangement IASI.FLOW). S. Whitburn is grateful for his Ph.D. grant (Boursier FRIA) to the "Fonds pour la Formation à la Recherche dans l'Industrie et dans l'Agriculture" of Belgium. L. Clarisse is Research Associate (Chercheur Qualifié) with the Belgian F.R.S.-FNRS. C. Clerbaux is grateful to CNES for scientific collaboration and financial support.





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





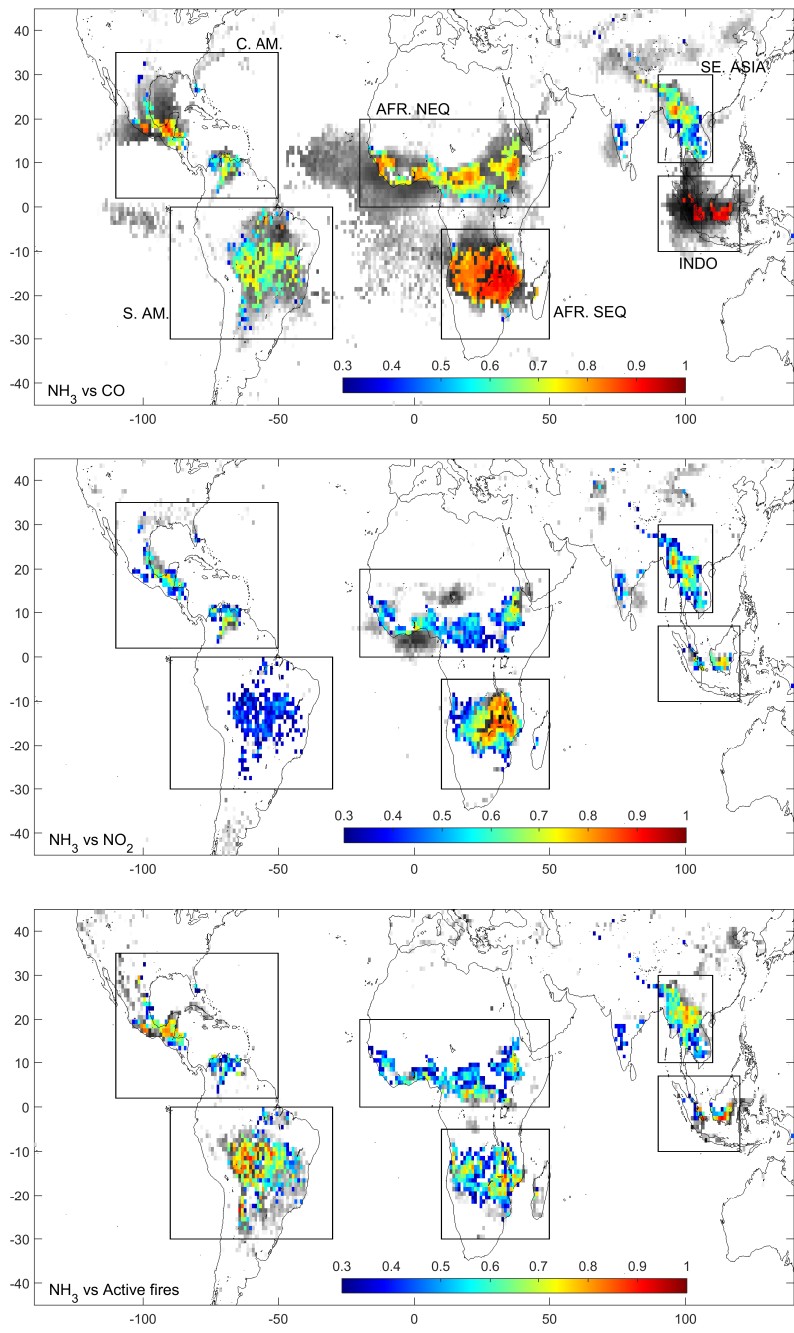

**Figure 1.** Correlation coefficients (r) of the linear regression of the monthly mean $NH_3$ total columns (molec.cm$^{-2}$) versus (a) CO total columns (molec.cm$^{-2}$) (top), (b) $NO_2$ total columns (molec.cm$^{-2}$) (middle) and (c) the number of active fires (bottom) from 2008 to 2015 in $1° \times 1°$ cells. Only pixels with a correlation coefficient r higher than 0.3 are shown. Pixels with r>0.3 for the three couples of regression are shown in color. Pixels with r>0.3 for the considered couple but not for (at least) one of the two other couples are shown in gray. The six regions selected for the study (C.AM., S.AM., AFR.NEQ., AFR.SEQ., SE.ASIA, INDO.) are highlighted by black rectangles





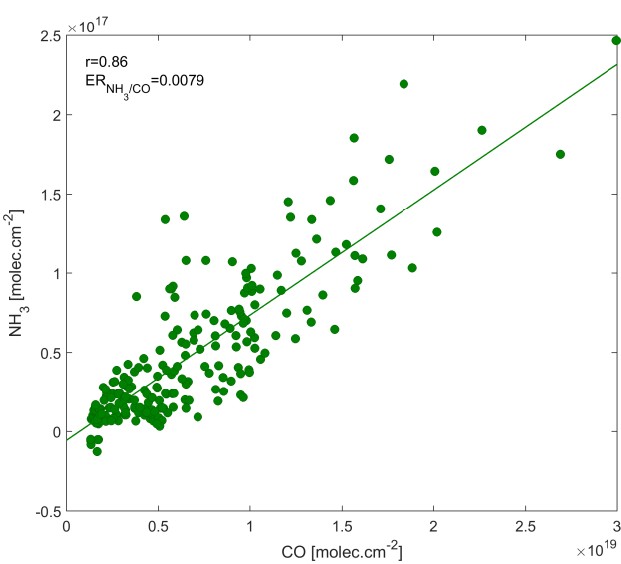

**Figure 2.** Example of linear regression between $NH_3$ and CO total columns (molec.cm$^{-2}$) for 2015 for one pixel of the selected grid-box, corresponding to the Evergreen Broadleaf Forest (EBF) biome in Indonesia (latitude=-3 deg; longitude=113 deg). The correlation coefficient (r) and the $ER_{NH_3/CO}$ (slope of the linear regression) are given as well as inset





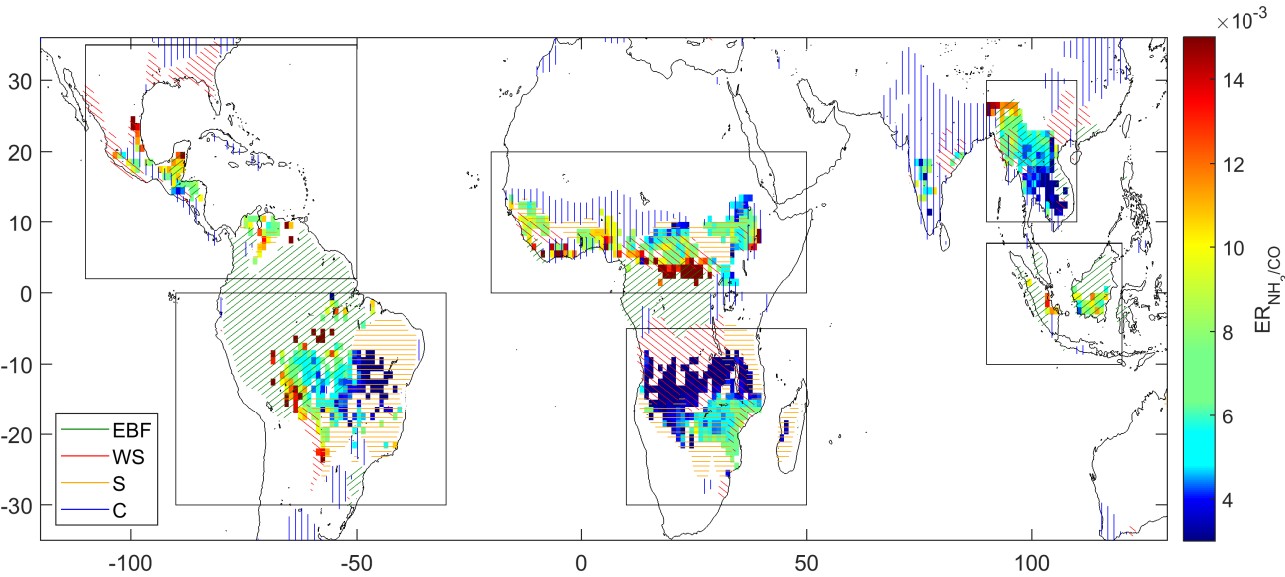

**Figure 3.** Mean yearly $ER_{NH_3/CO}$ averaged over the time period 2008 and 2015 for the selected pixels. The four main biomes studied are represented by the hatched lines: Savannas (S), Woody Savannas (WS), Evergreen Broadleaf Forest (EBF) and Crop together with the Crop and Natural Vegetation Mosaic (C+CNVM) here named as C





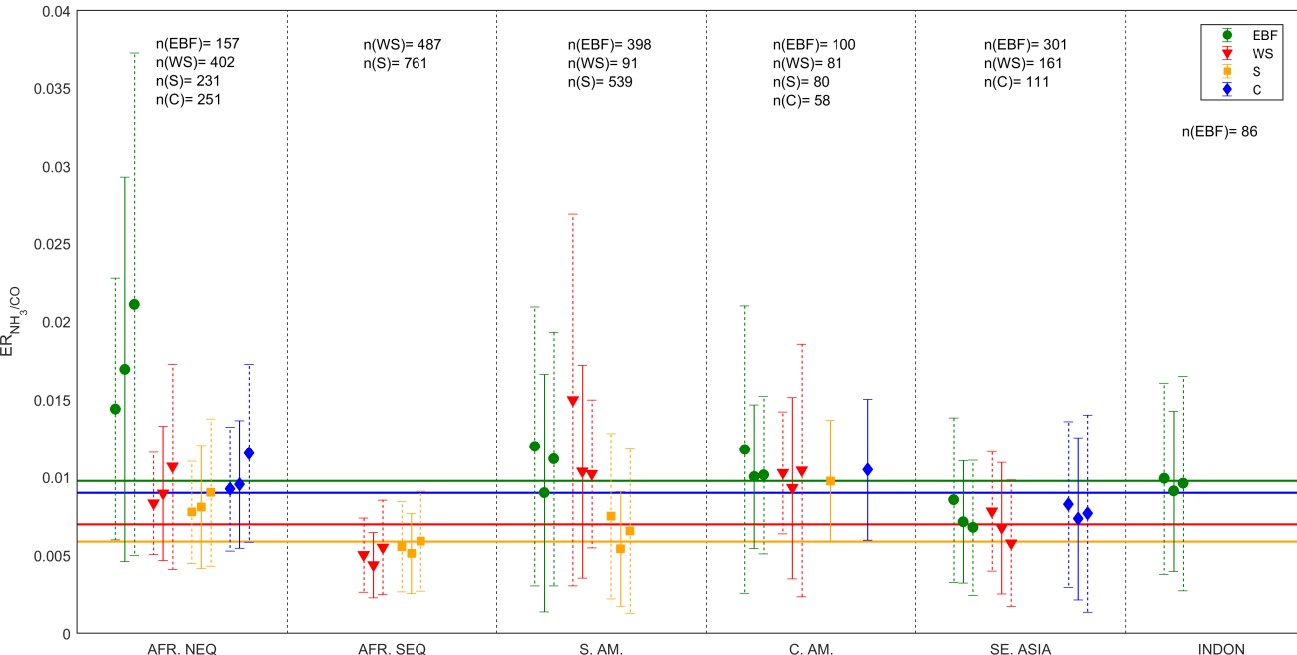

**Figure 4.** Mean $ER_{NH_3/CO}$ averaged for the six regions and four biomes from the yearly $ER_{NH_3/CO}$ (solid error bars) and from the early- and late fire season $ER_{NH_3/CO}$ (left and right dashed error bars, respectively) calculated between 2008 and 2015 for the pixels selected in section 2.3. The error bar is the 1-sigma standard deviation around the mean. The mean yearly $ER_{NH_3/CO}$ for each biome averaged globally for the six regions are indicated by the horizontal lines. n(x) (with x the biome) corresponds to the number of $ER_{NH_3/CO}$ averaged for each biome and region. Different symbols and colors are used for the different biomes. For the S and C biomes in the C.AM. region, no seasonal $ER_{NH_3/CO}$ are shown because of the lack of measurements





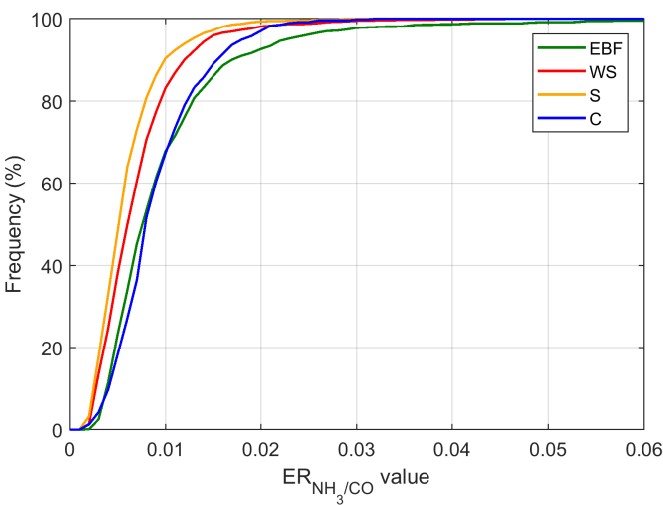

**Figure 5.** Cumulative curve of the yearly $ER_{NH_3/CO}$ calculated between 2008 and 2015 for the pixels selected in section 2.3 separated by biome



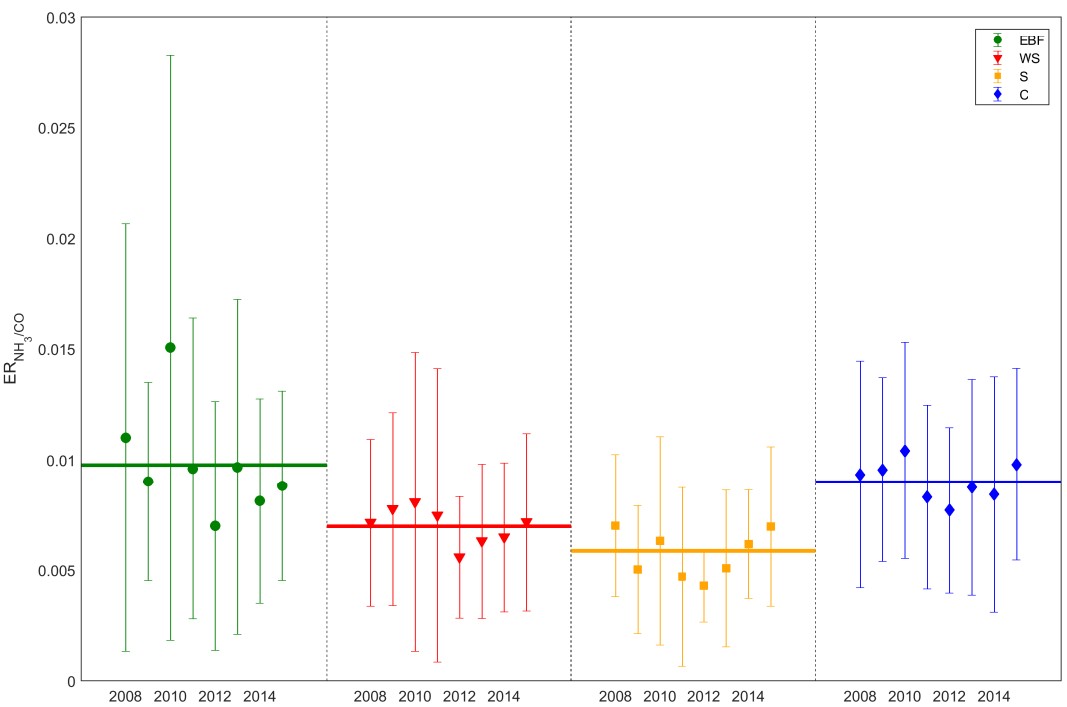

**Figure 6.** Mean $ER_{NH_3/CO}$ averaged by biome and by year (2008–2015) from the yearly $ER_{NH_3/CO}$ calculated for the pixels selected in section 2.3. The error bar is the 1-sigma standard deviation around the mean. The solid line represents the 8-years average per biome



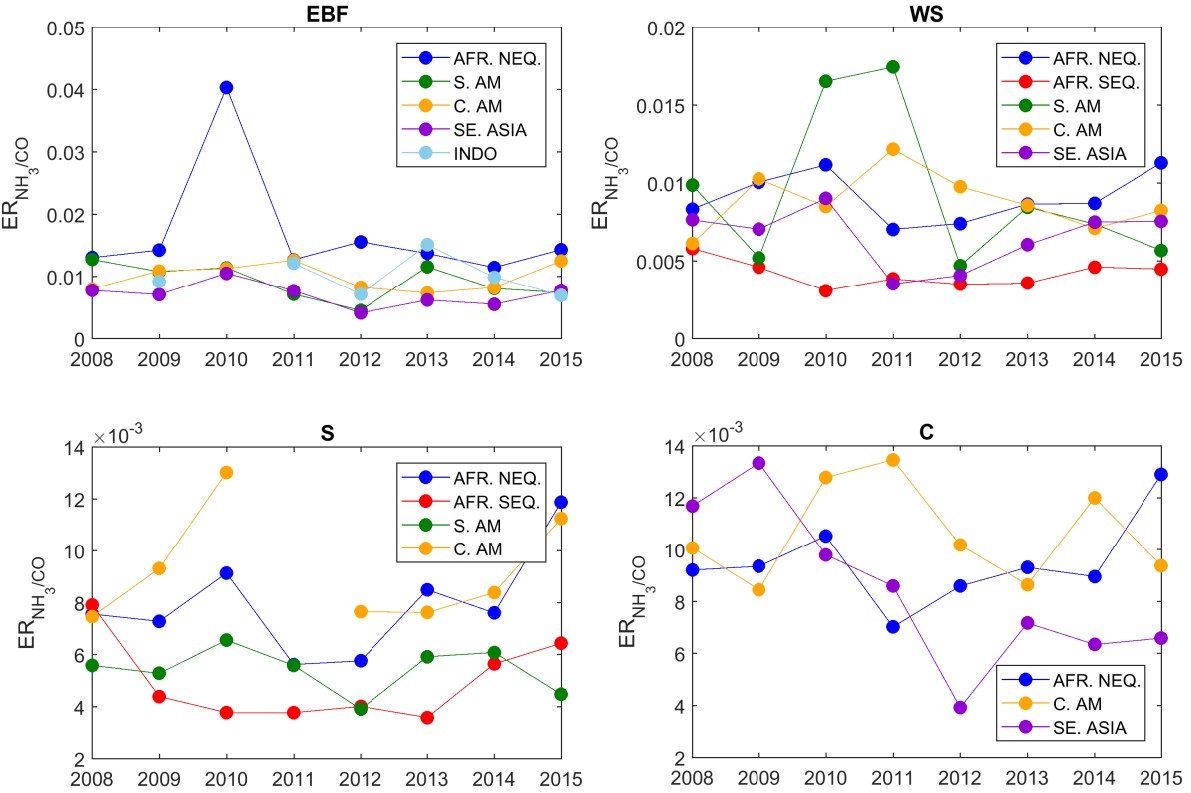

**Figure 7.** Mean biome-specific $ER_{NH_3/CO}$ averaged by year (2008–2015) and by region (colored dots and lines) from the yearly $ER_{NH_3/CO}$ calculated for the pixels selected in section 2.3. From top-left to bottom-right: EBF, WS, S and C



**Table 1.** $ER_{NH_3/CO}$ reported in the literature for different regions and biomes

| Source | NC Africa[a] | SC Africa[b] | S America[c] |
|---|---|---|---|
| **Luo et al. (2015) - TES** | $14\text{-}23\times10^{-3}$ | $-^d$ | $15\times10^{-3}$ |
| **Luo et al. (2015) - GEOS-Chem** | $8\text{-}17\times10^{-3}$ | $14\text{-}16\times10^{-3}$ | $11\times10^{-3}$ |
| **Source** | **Savanna** | **Tropical Forest** | **Cropland** |
| **Andreae and Merlet (2001)** | $15.2\times10^{-3}$ | $20.5\times10^{-3}$ | $23.3\times10^{-3}$ |
| **Bertschi et al. (2003)[e]** | $19.9\times10^{-3}$ | - | - |
| **Sinha et al. (2003)** | $7\times10^{-3}$ | - | - |
| **Yokelson et al. (2003)** | $6.5\text{-}7\times10^{-3}$ | - | - |
| **Christian et al. (2007)** | $12.8\times10^{-3}$ | - | - |
| **Akagi et al. (2011)** | $13.6\times10^{-3}$ | $23.6\times10^{-3}$ | $35.0\times10^{-3}$ |
| **Wooster et al. (2011)** | $8\text{-}35\times10^{-3}$ | - | - |
| **Yokelson et al. (2011)** | $9.9\times10^{-3}$ | $46.8\times10^{-3}$ [f] | $29.1\times10^{-3}$ |
| **Kaiser et al. (2012)** | $24.3\times10^{-3}$ | $15.2\times10^{-3}$ | $28.6\times10^{-3}$ |
| **Smith et al. (2014)** | $13.3\times10^{-3}$ | - | - |

[a,b,c] NC Africa=North-Central Africa, SC Africa=South-Central Africa, S America=South America

[d] too low correlation coefficient

[e] for smoldering logs

[f] Tropical dry forest