# Peer review of "IASI-derived NH3 enhancement ratios relative to CO for the tropical biomass burning regions"

_Atmospheric Chemistry and Physics, 2017_

## Referee Comment (RC1) · Anonymous Referee #1 · 4 Jun 2017

The manuscript by Whitburn et al. presents a new dataset of global enhancement ratios (ERs) of ammonia (NH3) relative to carbon monoxide (CO) for biomass burning events derived from multi-year observations of the IASI instrumentation on Metop satellites. The dataset is derived in combination with information on fires by the MODIS instrument and nitrogen dioxide (NO2) observations by GOME-2 on Metop. The calculated ERs are compared to a variety of previous measurements and are discussed in relation to different biomes and different regions. Further, their inter-annual and seasonal variability is investigated.

General comments:

[Figure]

This new dataset is very well presented and analysed and the methods are sufficiently described. The differences to previous observations are discussed with the necessary caution towards the own dataset and possible explanations are given. Unfortunately no clear cause for the discrepancies has been discovered which is not surprising given the large global variability and the various observational methods. In my opinion, the observational constraints on the NH3 and CO dataset should be a bit more detailed such that some estimated error range could be attributed to the derived ERs (see comments below). In summary, after having addressed the specific comments below, I support publication of the manuscript.

Specific comments:

3_30, 'have shown fair agreements between IASI-NH3 observations and other measurements':

The expression 'fair' is only qualitative. Please describe the agreement in more quantitative terms.

4_1, 'We also have assumed a similar sensitivity for IASI to NH3 and CO in the lower layers of the atmosphere, which is not expected to introduce a significant bias in the studied regions due to a generally positive thermal contrast prevailing during daytime':

- Can you demonstrate this assumption e.g. by presenting typical averaging kernels showing such a similar sensitivity of the NH3 and CO retrievals at the lower altitudes.

- Due to the different values of thermal contrast between surface and the lower atmosphere for different observations, an error could be introduced in the dataset which might partly be responsible for the overall variability of the ERs. Have you tried to correlate the temperature contrast with your dataset and can you exclude such an influence?

4_5, 'Whitburn et al. (2016b) have calculated that the use of an alternative profile could affect the retrieved column up to 50%':

[Figure]

It would be very helpful if one could try to derive a quantitative error assessment for the NH3/CO ratios combining estimated errors of NH3 and CO, taking into account such effects as described. At least it would be very helpful to put together a table where specific uncertainties and their sources are listed for NH3 and CO separately.

5_26, 'with a relative error lower than 100% for NH3':

Which kind of error is this? Only due to spectral noise?

8_5, 'The agreement would become even better if we consider in addition a possible bias due to the use of a non-representative NH3 vertical profile':

What is the reason that the agreement would improve and not becoming worse when using another a-priori NH3 profile?

10_8, 'These differences may be explained by various factors including. . .':

In this list, possible errors regarding the NH3 (and CO) IASI retrievals is missing, though it has been mentioned before.

23_Table1:

It would be very helpful if you could list, e.g. after each number, the result from the actual analysis for a similar biome and geographic region.

Technical comments:

3_29, 'on a two-dimensional look-up tables':

'on two-dimensional look-up tables'

8_27, 'are':

-> 'is'

---

## Referee Comment (RC2) · Anonymous Referee #2 · 1 Jul 2017

The paper by Whitburn is interesting and well written. I recommend publications.

I find myself asking two questions. I leave it to the authors to decide if these are within the scope of the paper.

1) Flaming vs. smoldering fires might have quite different CO but similar NH3. Is anything known? The paper gives the impression that CO is more constant than NH3. I don't know if that is true, but I think the paper should comment on whether variability is driven more by CO changes or more by the N-content of fuels. For example, wood has zero N, a recent paper Coggon et al. Geophys. Res. Lett. 2016 pointed out that residential wood burning has near zero emissions of N-compounds.

[Figure]

2) The relationship to NO2. Are the NH3 and NO2 columns related?

---

## Author Comment (AC1) · 7 Aug 2017

We would like to thank the referee for his/her positive feedback on the paper and useful comments. All have been addressed. The point-by-point responses to each of the comments are provided below. A 'tracked changes' version of the manuscript is also appended.

We would also like to draw your attention to the following. After the initial submission of the manuscript, a major update in the IASI-NH3 retrieval product was introduced. This update is of importance especially since we identified in the previous version of the NH3 dataset sharp discontinuities in the NH3 time series related to inconsisten-

cies between the different versions of the operational Eumetsat IASI L2 algorithms for temperature and clouds (which are used as input in the neural network). To tackle this issue, we have developed a version of the IASI-NH3 Neural Network which relies on the ERA-Interim ECMWF meteorological input data (along with built-in surface temperature) rather than on the Eumetsat IASI L2 data. This provides a reanalyzed dataset which is coherent in time over the whole period covered by IASI (2008-today) and can therefore be used to investigate interannual variability. The reanalyzed dataset has been described in a paper recently published in AMTD:

Van Damme, M., Whitburn, S., Clarisse, L., Clerbaux, C., Hurtmans, D., and Coheur, P.-F.: Version 2 of the IASI NH3 neural network retrieval algorithm; near-real time and reanalysed datasets, Atmos. Meas. Tech. Discuss., https://doi.org/10.5194/amt-2017-239, in review, 2017.

After careful analysis, we have judged that this new dataset should ideally be used also for the present manuscript. We have updated all the figures of the manuscript according to this new dataset. As you will see, the main conclusions of our paper remain unchanged, except for the comparison with the ERNH3/CO from the literature. The IASI derived ERNH3/CO are higher than in the manuscript initially submitted and this leads to a much better agreement with the values given in the literature. With these changes and the comments addressed, we are confident that the paper has been greatly improved.

———————-

3_30: The expression 'fair' is only qualitative. Please describe the agreement in more quantitative terms.

We have changed the sentence in the manuscript to:

Two studies, based on a previous NH3 retrieval algorithm also using the HRI but relying on two-dimensional look-up tables for the conversion into a NH3 total column

(molec.cm-2) (Van Damme et al., 2014), have shown a fair agreement between IASI-NH3 observations and other measurements (generally within the uncertainties of the IASI NH3 retrieved columns), with differences of about 60-80% reported in Van Damme et al. (2015) and of 30% on average in Dammers et al. (2016).

4_1: "We also have assumed a similar sensitivity for IASI to NH3 and CO in the lower layers of the atmosphere, which is not expected to introduce a significant bias in the studied regions due to a generally positive thermal contrast prevailing during daytime". - Can you demonstrate this assumption e.g. by presenting typical averaging kernels showing such a similar sensitivity of the NH3 and CO retrievals at the lower altitudes?

The retrieval algorithm for NH3 does not provide averaging kernels. It is therefore difficult to compare quantitatively the sensitivity of NH3 and CO in the lower layers. However, the sensitivity of thermal nadir measurements near the surface is intimately related to the thermal contrast between the surface and the first layers of the atmosphere (Clerbaux et al., 2009). For NH3, Clarisse et al. (2010) have shown that, when the detection is possible (i.e. in case of good thermal contrast), the peak sensitivity for NH3 is in the boundary layer. For CO, Georges et al. (2009) and Bauduin et al. (2016) have also demonstrated the good sensitivity of IASI in the lower layers in case of high thermal contrast. Since we consider here only daytime measurements, in tropical regions and with a retrieval uncertainty on the NH3 column lower than 100% (i.e. a good sensitivity), the majority of the observations retained correspond to this situation of good thermal contrast.

George, M., Clerbaux, C., Hurtmans, D., Turquety, S., Coheur, P.-F., Pommier, M., Hadji-Lazaro, J., Edwards, D. P., Worden, H., Luo, M., Rinsland, C., and McMillan, W. (2009). Carbon monoxide distributions from the IASI/METOP mission: evaluation with other space-borne remote sensors. Atmos. Chem. Phys., 9(21):8317–8330. doi: 10.5194/acp-9-8317-2009.

Bauduin, S. and Clarisse, L. and Theunissen, M. and George, M. and Hurtmans, D. and

Clerbaux, C. and Coheur, P. F. (2016). IASI's sensitivity to near-surface carbon monoxide (CO): Theoretical analyses and retrievals on test cases. Journal of Quantitative Spectroscopy & Radiative Transfer, 189:428-440. doi: /10.1016/j.jqsrt.2016.12.022.

To clarify this, we have adapted the following sentence in the manuscript to:

We also have assumed a similar sensitivity for IASI to NH3 and CO in the lower layers of the atmosphere. This is not expected to introduce a significant bias since it has been shown for both CO and NH3 that the peak sensitivity was in the lower layers of the atmosphere in case of positive thermal contrast which is generally prevailing in the studied regions during daytime (George et al., 2009; Clarisse et al., 2010; Van Damme et al., 2014; Bauduin et al., 2016).

- Due to the different values of thermal contrast between surface and the lower atmosphere for different observations, an error could be introduced in the dataset which might partly be responsible for the overall variability of the ERs. Have you tried to correlate the temperature contrast with your dataset and can you exclude such an influence?

It is true that the thermal contrast greatly influence the sensitivity of IASI to near-surface measurements. However, as mentioned here above, for the observation considered, we are generally in a situation of high positive thermal contrast where NH3 (and CO) can generally be retrieved with a good accuracy. Moreover, the sensitivity of the retrieved column to the atmospheric parameters (including the thermal contrast) is included in the uncertainty.

4_5: "Whitburn et al. (2016b) have calculated that the use of an alternative profile could affect the retrieved column up to 50%" It would be very helpful if one could try to derive a quantitative error assessment for the NH3/CO ratios combining estimated errors of NH3 and CO, taking into account such effect as described. At least it would be very helpful to put together a table where specific uncertainties and their source are listed for NH3 and CO separately.

Since the ERNH3/CO are calculated from the slope of the linear regression of NH3 versus CO total columns, they do not rely on the individual uncertainties of each NH3 and CO observation. It is therefore very difficult to estimate a quantitative error assessment for the NH3/CO ratios. As explained in the next comment, the total uncertainty on the NH3 retrieved columns depend on the atmospheric conditions and on the amount of NH3. Examples of the relative importance of the different atmospheric parameters on the total uncertainty are given in Whitburn et al. (2016b). The observations considered in this study correspond generally to a situation of high thermal contrast and a relatively high NH3 column. In the case of high thermal contrast and high NH3 column, Whitburn et al. (2016b) have shown that the contribution to the total uncertainty was well distributed on the different parameters while, for high thermal contrast but lower columns, the major contribution to the total uncertainty was on the HRI (mainly instrumental noise). Note that Whitburn et al. (2016b) have shown that, despite a possible high uncertainty, the average bias on the columns is close to zero. For CO, the uncertainty is generally lower (of about 5-10% and up to 20% for the observations considered here). The uncertainty on the calculated ERNH3/CO is therefore mainly driven by the uncertainty on the NH3 column (which is lower than 100% due to the pre-filtering applied).

5_26: "With a relative error lower than 100% for NH3": Which kind of error is this? Only due to spectral noise?

The error associated with the NH3 retrieved columns includes a full uncertainty analysis and depend on the atmospheric conditions and the amount of NH3. The uncertainty analysis is performed by perturbing the input parameters (Temperature profile, NH3 a priori vertical profile, HRI, etc.) of the neural network and evaluates for each retrieved column how the uncertainty of the input parameters propagates to the final result. We have added the following sentence in the manuscript to make it more clear for the reader (3_21):

The retrieval also includes a full uncertainty analysis, performed by perturbing the input

parameters (temperature profile, HRI, NH3 a priori profile, etc.) of the neural network.

8_5: "The agreement would become even better if we consider in addition a possible bias due to the use of a non-representative NH3 vertical profile." What is the reason that the agreement would improve and not becoming worse when using another a-priori NH3 profile?

Thank you for pointing this out. It is indeed very hard to evaluate whether the 'real' NH3 profiles would increase or decrease the retrieved columns because of the large variety of NH3 profiles associated with transported fire plumes. As mentioned above, Clarisse et al. (2010) have shown that in case of favorable thermal contrast, the peak sensitivity of IASI for NH3 is in the boundary layer (below 1 km). For the fire plumes, Val Martin et al. (2010) have shown that, for tropical forests and grasslands/croplands fires, the plume height was generally below the boundary layer height while only a small fraction peaked above 1.5 km. Due to a lack of available information on plume height for each individual fires, it is hard to determine the impact of the true profile on the retrieved columns (and therefore on the ratio NH3/CO).

M. Val Martin, J. A. Logan, R. A. Kahn, F.-Y. Leung, D. L. Nelson, and D. J. Diner. Smoke injection heights from fires in North America: analysis of 5 years of satellite observations. Atmospheric Chemistry and Physics, 10(4):1491-1510, 2010. doi: 10.5194/acp-10-1491-2010.

We have adapted the following sentence to clarify this:

Finally, as we mentioned in section 2.1, the retrieval of NH3 could be biased by the use of a constant NH3 vertical profile not representative of the variety of profiles observed above biomass burning plumes.

10_8: "These differences may be explained by various factors including...": In the list, possible errors regarding the NH3 (and CO) IASI retrievals is missing though it has been mentioned before.

We have modified the following sentence to take this into account:

This may be explained by various factors including 1) the parametrization (pre- and post-filtering of the data) considered for the calculation of the ERNH3/CO, 2) a bias towards the flaming phase due to the selection of IASI observations close to MODIS active fires (less sensitive to the smoldering phase) and 3) a possible accumulation of CO in the region during the fire season, introducing a low-bias in the IASI-derived ERNH3/CO. Another possible explanation might lie in the use of a unique vertical profile shape in the retrieval scheme of NH3 while biomass burning plumes exhibit a large variety of plume injection heights.

23_Table1: It would be very helpful if you could list, e.g. after each number, the result from the actual analysis for a similar biome and geographic region.

This is indeed a good suggestion. We have added to table 1 two rows including the results from the actual analysis (one on a regional level and one for a biome level).

Technical comments have been taken into account.

Please also note the supplement to this comment:
https://www.atmos-chem-phys-discuss.net/acp-2017-331/acp-2017-331-AC1-supplement.pdf

―――――――――――――――――――

---

## Author Comment (AC2) · 7 Aug 2017

We would like to thank the referee for his/her positive feedback on the paper and useful comments. All have been addressed. The point-by-point responses to each of the comments are provided below. A 'tracked changes' version of the manuscript is also appended.

We would also like to draw the attention of the referee to the following. After the initial submission of the manuscript, a major update in the IASI-NH3 retrieval product was introduced. This update is of importance especially since we identified in the previous version of the NH3 dataset sharp discontinuities in the NH3 time series related

to inconsistencies between the different versions of the operational Eumetsat IASI L2 algorithms for temperature and clouds (which are used as input in the neural network). To tackle this issue, we have developed a version of the IASI-NH3 Neural Network which relies on the ERA-Interim ECMWF meteorological input data (along with built-in surface temperature) rather than on the Eumetsat IASI L2 data. This provides a reanalyzed dataset which is coherent in time over the whole period covered by IASI (2008-today) and can therefore be used to investigate interannual variability. The reanalyzed dataset has been described in a paper recently published in AMTD:

Van Damme, M., Whitburn, S., Clarisse, L., Clerbaux, C., Hurtmans, D., and Coheur, P.-F.: Version 2 of the IASI NH3 neural network retrieval algorithm; near-real time and reanalysed datasets, Atmos. Meas. Tech. Discuss., https://doi.org/10.5194/amt-2017-239, in review, 2017.

After careful analysis, we have judged that this new dataset should ideally be used also for the present manuscript. We have updated all the figures of the manuscript according to this new dataset. As you will see, the main conclusions of our paper remain unchanged, except for the comparison with the ERNH3/CO from the literature. The IASI derived ERNH3/CO are higher than in the manuscript initially submitted and this leads to a much better agreement with the values given in the literature. With these changes and the comments addressed, we are confident that the paper has been greatly improved.
* * *
The relationship to NO2. Are the NH3 and NO2 columns related?

NO2 and NH3 emissions are anti-correlated during the different stages of a fire. During the flaming phase, which takes place first, emissions of oxidized species (including NO2) dominate. When all the easily accessible fuel has been consumed, smoldering combustion starts with increased emissions of reduced or incompletely oxidized species (NH3, CO, etc.) due to the lack of available atmospheric oxygen and/or the lower temperatures.

Flaming vs. smoldering fires might have quite different CO but similar NH3. Is anything known? The paper gives the impression that CO is more constant than NH3. I don't know if that is true, but I think the paper should comment on whether variability is driven more by CO changes or more by the N-content of fuels. For example, wood has zero N, a recent paper Coggon et al. Geophys. Res. Lett. 2016 pointed out that residential wood burning has near zero emissions of N-compounds.

Flaming and smoldering phases show very different proportions of NH3 emissions relative to the total Nr emissions. As mentioned above, emissions of oxidized Nr species (mainly NO2) dominate during the flaming phase while, during the smoldering phase, NH3 becomes the dominant Nr species emitted. CO also dominates during the smoldering phase.

This is indeed true (and well known) that flaming and smoldering phases will have very different ERNH3/CO. This is observed in particular in this study when looking at the differences between the ERNH3/CO for tropical forest and the savanna fires. Forest fires, which are characterized by a larger fraction of smoldering combustion than savannas (due to the higher biomass density) show higher ERNH3/CO than savannas. When studying enhancement ratios from satellite observations, we calculate an average ERNH3/CO (of generally an entire fire) over a certain time period and area which therefore include both smoldering and fire phases. Resulting average ERNH3/CO is therefore the result of many different factors acting together. It is however not possible to separate the contribution of each ones.

Please also note the supplement to this comment:
https://www.atmos-chem-phys-discuss.net/acp-2017-331/acp-2017-331-AC2-supplement.pdf

―――――――――――――――――

2017.

**Supplement:**

[revised manuscript text omitted]

---

## Author Response (AR1)

Dear Editor,

Please find enclosed the revised version of our manuscript entitled "IASI-derived $NH_3$ enhancement ratios relative to CO for the tropical biomass burning regions". We have addressed all the comments of the referees.

We would also like to draw your attention to the following. After the initial submission of the manuscript, a major update in the IASI-$NH_3$ retrieval product was introduced. This update is of importance especially since we identified in the previous version of the $NH_3$ dataset sharp discontinuities in the $NH_3$ time series related to inconsistencies between the different versions of the operational Eumetsat IASI L2 algorithms for temperature and clouds (which are used as input in the neural network). To tackle this issue, we have developed a version of the IASI-$NH_3$ Neural Network which relies on the ERA-Interim ECMWF meteorological input data (along with built-in surface temperature) rather than on the Eumetsat IASI L2 data. This provides a reanalyzed dataset which is coherent in time over the whole period covered by IASI (2008-today) and can therefore be used to investigate interannual variability. The reanalyzed dataset has been described in a paper recently published in AMTD[1].

After careful analysis, we have judged that this new dataset should ideally be used also for the present manuscript. We have updated all the figures of the manuscript according to this new dataset. As you will see, the main conclusions of our paper remain unchanged, except for the comparison with the $ER_{NH3/CO}$ from the literature. The IASI derived $ER_{NH3/CO}$ are higher than in the manuscript initially submitted and this leads to a much better agreement with the values given in the literature. With these changes and the comments of the referees addressed, we are confident that the paper has been greatly improved.

The responses to each of the referee's comments are provided here below. A 'tracked changes' version of the manuscript is appended and allows to better identify where and how the manuscript was modified. We hope that with these changes and the replies made to the referees, you will find the manuscript suitable for publication in ACP.

Yours sincerely,

Simon Whitburn.
* * *
[1] Van Damme, M., Whitburn, S., Clarisse, L., Clerbaux, C., Hurtmans, D., and Coheur, P.-F.: Version 2 of the IASI NH3 neural network retrieval algorithm; near-real time and reanalysed datasets, Atmos. Meas. Tech. Discuss., https://doi.org/10.5194/amt-2017-239, in review, 2017.

**Response to reviewers:**

Reviewer #1:

**We would like to thank the reviewer for his/her positive feedback on the paper and useful comments. All have been addressed. The point-by-point responses to each of the comments are provided below. A 'tracked changes' version of the manuscript is also appended.**

**We would also like to draw your attention to the following. After the initial submission of the manuscript, a major update in the IASI-NH$_3$ retrieval product was introduced. This update is of importance especially since we identified in the previous version of the NH$_3$ dataset sharp discontinuities in the NH$_3$ time series related to inconsistencies between the different versions of the operational Eumetsat IASI L2 algorithms for temperature and clouds (which are used as input in the neural network). To tackle this issue, we have developed a version of the IASI-NH$_3$ Neural Network which relies on the ERA-Interim ECMWF meteorological input data (along with built-in surface temperature) rather than on the Eumetsat IASI L2 data. This provides a reanalyzed dataset which is coherent in time over the whole period covered by IASI (2008-today) and can therefore be used to investigate interannual variability. The reanalyzed dataset has been described in a paper recently published in AMTD[2].**

**After careful analysis, we have judged that this new dataset should ideally be used also for the present manuscript. We have updated all the figures of the manuscript according to this new dataset. As you will see, the main conclusions of our paper remain unchanged, except for the comparison with the ER$_{NH3/CO}$ from the literature. The IASI derived ER$_{NH3/CO}$ are higher than in the manuscript initially submitted and this leads to a much better agreement with the values given in the literature. With these changes and the comments addressed, we are confident that the paper has been greatly improved.**

3_30: The expression 'fair' is only qualitative. Please describe the agreement in more quantitative terms.

**We have changed the sentence in the manuscript to:**
Two studies, based on a previous NH$_3$ retrieval algorithm also using the HRI but relying on two-dimensional look-up tables for the conversion into a NH$_3$ total column (molec.cm$^{-2}$) (Van Damme et al., 2014), have shown a fair agreement between IASI-NH$_3$ observations and other measurements (generally within the uncertainties of the IASI NH$_3$ retrieved columns), with differences of about 60-80% reported in Van Damme et al. (2015) and of 30% on average in Dammers et al. (2016).
* * *
[2] Van Damme, M., Whitburn, S., Clarisse, L., Clerbaux, C., Hurtmans, D., and Coheur, P.-F.: Version 2 of the IASI NH3 neural network retrieval algorithm; near-real time and reanalysed datasets, Atmos. Meas. Tech. Discuss., https://doi.org/10.5194/amt-2017-239, in review, 2017.

4_1: "We also have assumed a similar sensitivity for IASI to NH3 and CO in the lower layers of the atmosphere, which is not expected to introduce a significant bias in the studied regions due to a generally positive thermal contrast prevailing during daytime".

− Can you demonstrate this assumption e.g. by presenting typical averaging kernels showing such a similar sensitivity of the NH3 and CO retrievals at the lower altitudes?

**The retrieval algorithm for $NH_3$ does not provide averaging kernels. It is therefore difficult to compare quantitatively the sensitivity of $NH_3$ and CO in the lower layers. However, the sensitivity of thermal nadir measurements near the surface is intimately related to the thermal contrast between the surface and the first layers of the atmosphere (Clerbaux et al., 2009). For $NH_3$, Clarisse et al. (2010) have shown that, when the detection is possible (i.e. in case of good thermal contrast), the peak sensitivity for $NH_3$ is in the boundary layer. For CO, Georges et al. (2009) and Bauduin et al. (2016) have also demonstrated the good sensitivity of IASI in the lower layers in case of high thermal contrast. Since we consider here only daytime measurements, in tropical regions and with a retrieval uncertainty on the $NH_3$ column lower than 100% (i.e. a good sensitivity), the majority of the observations retained correspond to this situation of good thermal contrast.**

**George, M., Clerbaux, C., Hurtmans, D., Turquety, S., Coheur, P.-F., Pommier, M., Hadji-Lazaro, J., Edwards, D. P., Worden, H., Luo, M., Rinsland, C., and McMillan, W. (2009). Carbon monoxide distributions from the IASI/METOP mission: evaluation with other space-borne remote sensors. Atmos. Chem. Phys., 9(21):8317–8330. doi: 10.5194/acp-9-8317-2009.**

**Bauduin, S. and Clarisse, L. and Theunissen, M. and George, M. and Hurtmans, D. and Clerbaux, C. and Coheur, P. F. (2016). IASI's sensitivity to near-surface carbon monoxide (CO): Theoretical analyses and retrievals on test cases. Journal of Quantitative Spectroscopy & Radiative Transfer, 189:428-440. doi: /10.1016/j.jqsrt.2016.12.022.**

**To clarify this, we have adapted the following sentence in the manuscript to:**

**We also have assumed a similar sensitivity for IASI to $NH_3$ and CO in the lower layers of the atmosphere. This is not expected to introduce a significant bias since it has been shown for both CO and $NH_3$ that the peak sensitivity was in the lower layers of the atmosphere in case of positive thermal contrast which is generally prevailing in the studied regions during daytime (George et al., 2009; Clarisse et al., 2010; Van Damme et al., 2014; Bauduin et al., 2016).**

− Due to the different values of thermal contrast between surface and the lower atmosphere for different observations, an error could be introduced in the dataset which might partly be responsible for the overall variability of the ERs. Have you tried to correlate the temperature contrast with your dataset and can you exclude such an influence?

**It is true that the thermal contrast greatly influence the sensitivity of IASI to near-surface measurements. However, as mentioned here above, for the observation considered, we are generally in a situation of high positive thermal contrast where NH₃ (and CO) can generally be retrieved with a good accuracy. Moreover, the sensitivity of the retrieved column to the atmospheric parameters (including the thermal contrast) is included in the uncertainty.**

4_5: "Whitburn et al. (2016b) have calculated that the use of an alternative profile could affect the retrieved column up to 50%" It would be very helpful if one could try to derive a quantitative error assessment for the NH3/CO ratios combining estimated errors of NH3 and CO, taking into account such effect as described. At least it would be very helpful to put together a table where specific uncertainties and their source are listed for NH3 and CO separately.

**Since the $ER_{NH3/CO}$ are calculated from the slope of the linear regression of NH₃ versus CO total columns, they do not rely on the individual uncertainties of each NH₃ and CO observation. It is therefore very difficult to estimate a quantitative error assessment for the NH₃/CO ratios.**
**As explained in the next comment, the total uncertainty on the NH₃ retrieved columns depend on the atmospheric conditions and on the amount of NH₃. Examples of the relative importance of the different atmospheric parameters on the total uncertainty are given in Whitburn et al. (2016b). The observations considered in this study correspond generally to a situation of high thermal contrast and a relatively high NH₃ column. In the case of high thermal contrast and high NH₃ column, Whitburn et al. (2016b) have shown that the contribution to the total uncertainty was well distributed on the different parameters while, for high thermal contrast but lower columns, the major contribution to the total uncertainty was on the HRI (mainly instrumental noise). Note that Whitburn et al. (2016b) have shown that, despite a possible high uncertainty, the average bias on the columns is close to zero.**
**For CO, the uncertainty is generally lower (of about 5-10% and up to 20% for the observations considered here). The uncertainty on the calculated $ER_{NH3/CO}$ is therefore mainly driven by the uncertainty on the NH₃ column (which is lower than 100% due to the pre-filtering applied).**

5_26: "With a relative error lower than 100% for NH3": Which kind of error is this? Only due to spectral noise?

**The error associated with the NH₃ retrieved columns includes a full uncertainty analysis and depend on the atmospheric conditions and the amount of NH₃. The uncertainty analysis is performed by perturbing the input parameters (Temperature profile, NH₃ a priori vertical profile, HRI, etc.) of the neural network and evaluates for each retrieved column how the uncertainty of the input parameters propagates to the final result.**

**We have added the following sentence in the manuscript to make it more clear for the reader (3_21):**

**The retrieval also includes a full uncertainty analysis, performed by perturbing the input parameters (temperature profile, HRI, NH₃ a priori profile, etc.) of the neural network.**

8_5: "The agreement would become even better if we consider in addition a possible bias due to the use of a non-representative NH₃ vertical profile." What is the reason that the agreement would improve and not becoming worse when using another a-priori NH₃ profile?

**Thank you for pointing this out. It is indeed very hard to evaluate whether the 'real' NH₃ profiles would increase or decrease the retrieved columns because of the large variety of NH₃ profiles associated with transported fire plumes. As mentioned above, Clarisse et al. (2010) have shown that in case of favorable thermal contrast, the peak sensitivity of IASI for NH₃ is in the boundary layer (below 1 km). For the fire plumes, Val Martin et al. (2010) have shown that, for tropical forests and grasslands/croplands fires, the plume height was generally below the boundary layer height while only a small fraction peaked above 1.5 km. Due to a lack of available information on plume height for each individual fires, it is hard to determine the impact of the true profile on the retrieved columns (and therefore on the ratio NH₃/CO).**

**M. Val Martin, J. A. Logan, R. A. Kahn, F.-Y. Leung, D. L. Nelson, and D. J. Diner. Smoke injection heights from fires in North America: analysis of 5 years of satellite observations. Atmospheric Chemistry and Physics, 10(4):1491-1510, 2010. doi: 10.5194/acp-10-1491-2010.**

**We have adapted the following sentence to clarify this:**
**Finally, as we mentioned in section 2.1, the retrieval of NH₃ could be biased by the use of a constant NH₃ vertical profile not representative of the variety of profiles observed above biomass burning plumes.**

10_8: "These differences may be explained by various factors including…": In the list, possible errors regarding the NH3 (and CO) IASI retrievals is missing though it has been mentioned before.

**We have modified the following sentence to take this into account:**
**This may be explained by various factors including 1) the parametrization (pre- and post-filtering of the data) considered for the calculation of the $ER_{NH3/CO}$, 2) a bias towards the flaming phase due to the selection of IASI observations close to MODIS active fires (less sensitive to the smoldering phase) and 3) a possible accumulation of CO in the region during the fire season, introducing a low-bias in the IASI-derived $ER_{NH3/CO}$. Another possible explanation might lie in the use of a unique vertical profile shape in the retrieval scheme of NH₃ while biomass burning plumes exhibit a large variety of plume injection heights.**

23_Table1: It would be very helpful if you could list, e.g. after each number, the result from the actual analysis for a similar biome and geographic region.

**This is indeed a good suggestion. We have added to table 1 two rows including the results from the actual analysis (one on a regional level and one for a biome level).**

**Technical comments have been taken into account.**

Reviewer #2:

**We would like to thank the referee for his/her positive feedback on the paper and useful comments. All have been addressed. The point-by-point responses to each of the comments are provided below. A 'tracked changes' version of the manuscript is also appended.**

**We would also like to draw the attention of the referee to the following. After the initial submission of the manuscript, a major update in the IASI-NH$_3$ retrieval product was introduced. This update is of importance especially since we identified in the previous version of the NH$_3$ dataset sharp discontinuities in the NH$_3$ time series related to inconsistencies between the different versions of the operational Eumetsat IASI L2 algorithms for temperature and clouds (which are used as input in the neural network). To tackle this issue, we have developed a version of the IASI-NH$_3$ Neural Network which relies on the ERA-Interim ECMWF meteorological input data (along with built-in surface temperature) rather than on the Eumetsat IASI L2 data. This provides a reanalyzed dataset which is coherent in time over the whole period covered by IASI (2008-today) and can therefore be used to investigate interannual variability. The reanalyzed dataset has been described in a paper recently published in AMTD[3].**

**After careful analysis, we have judged that this new dataset should ideally be used also for the present manuscript. We have updated all the figures of the manuscript according to this new dataset. As you will see, the main conclusions of our paper remain unchanged, except for the comparison with the ER$_{NH3/CO}$ from the literature. The IASI derived ER$_{NH3/CO}$ are higher than in the manuscript initially submitted and this leads to a much better agreement with the values given in the literature. With these changes and the comments addressed, we are confident that the paper has been greatly improved.**

The relationship to NO$_2$. Are the NH$_3$ and NO$_2$ columns related?

**NO$_2$ and NH$_3$ emissions are anti-correlated during the different stages of a fire. During the flaming phase, which takes place first, emissions of oxidized species (including NO$_2$) dominate. When all the easily accessible fuel has been consumed, smoldering combustion starts with increased emissions of reduced or incompletely oxidized species (NH$_3$, CO, etc.) due to the lack of available atmospheric oxygen and/or the lower temperatures.**

Flaming vs. smoldering fires might have quite different CO but similar NH$_3$. Is anything known? The paper gives the impression that CO is more constant than NH$_3$. I don't know if
* * *
[3] Van Damme, M., Whitburn, S., Clarisse, L., Clerbaux, C., Hurtmans, D., and Coheur, P.-F.: Version 2 of the IASI NH3 neural network retrieval algorithm; near-real time and reanalysed datasets, Atmos. Meas. Tech. Discuss., https://doi.org/10.5194/amt-2017-239, in review, 2017.

that is true, but I think the paper should comment on whether variability is driven more by CO changes or more by the N-content of fuels. For example, wood has zero N, a recent paper Coggon et al. Geophys. Res. Lett. 2016 pointed out that residential wood burning has near zero emissions of N-compounds.

**Flaming and smoldering phases show very different proportions of $NH_3$ emissions relative to the total Nr emissions. As mentioned above, emissions of oxidized Nr species (mainly $NO_2$) dominate during the flaming phase while, during the smoldering phase, $NH_3$ becomes the dominant Nr species emitted. CO also dominates during the smoldering phase.**

**This is indeed true (and well known) that flaming and smoldering phases will have very different $ER_{NH3/CO}$. This is observed in particular in this study when looking at the differences between the $ER_{NH3/CO}$ for tropical forest and the savanna fires. Forest fires, which are characterized by a larger fraction of smoldering combustion than savannas (due to the higher biomass density) show higher $ER_{NH3/CO}$ than savannas. When studying enhancement ratios from satellite observations, we calculate an average $ER_{NH3/CO}$ (of generally an entire fire) over a certain time period and area which therefore include both smoldering and fire phases. Resulting average $ER_{NH3/CO}$ is therefore the result of many different factors acting together. It is however not possible to separate the contribution of each ones.**

[revised manuscript text omitted]